# The influence of patriarchy on Nepali-speaking Bhutanese women's diabetes self-management

**Aditi Sharma**[1]*, **Heather Stuckey**[2,3], **Megan Mendez-Miller**[4], **Yendelela Cuffee**[5], **Aubrey J. Juris**[6], **Jennifer S. McCall-Hosenfeld**[2,3]

**1** Department of Urology, School of Medicine, Stanford University, Stanford, California, United States of America, **2** Department of Public Health Sciences, College of Medicine, Pennsylvania State University, Hershey, Pennsylvania, United States of America, **3** Department of Medicine, College of Medicine, Pennsylvania State University, Hershey, Pennsylvania, United States of America, **4** Department of Family and Community Medicine, Pennsylvania State College of Medicine, Hershey, Pennsylvania, United States of America, **5** Program in Epidemiology, College of Health Sciences, University of Delaware, Newark, Delaware, United States of America, **6** Pennsylvania Department of Health Office of Health Equity, Harrisburg, Pennsylvania, United States of America

☯ These authors contributed equally to this work.

\* aditish@stanford.edu

**Data Availability Statement:** All relevant data are within the paper and its Supporting information files.

## Abstract

### Introduction

The Nepali-speaking Bhutanese (NSB) community is a rapidly growing population in Central Pennsylvania. A community-based diabetes education pilot program found a large gender disparity with fewer women in attendance; participants reported that primary household cooks and caretakers were women. This may be an indication of women's status in the NSB community, their healthcare access, autonomy, and ability to manage their diabetes. Hence, this study aims to understand the manifestations of patriarchy and its impact on NSB women's diabetes self-management employing a conceptual framework based on Walby's structures of patriarchy.

### Methods

An exploratory feminist qualitative inquiry was conducted. Fifteen NSB women with Type 2 Diabetes were interviewed about their diabetes self-management. Transcripts were coded for key concepts that emerged from the data. A thematic analysis was conducted. Themes were developed inductively through those categories as well as through an a priori approach using the conceptual framework.

### Results

Cultural influences such as family structure, religious beliefs, traditional healthcare and gender roles determined NSB women's patriarchal upbringing and lifestyle. Unpaid household production was largely dependent on women. Multiple immigrations led to poor socioeconomic indicators and marginalization of NSB women. Women's access to healthcare

**Funding:** The author(s) received no specific funding for this work.

**Competing interests:** The authors have declared that no competing interests exist.

(including diabetes) was entirely reliant on other family members due to poor autonomy. Women experienced adverse physical and emotional symptoms related to diabetes and their ability and attempts to maintain a healthy diabetes lifestyle was determined by their physical health condition, knowledge regarding good dietary practices and self-efficacy.

## Conclusion

Patriarchal practices that start early on within women's lives, such as child marriage, religious restrictions as well as women's access to education and autonomy impacted NSB women's access to healthcare, knowledge regarding their diabetes and self-efficacy. Future interventions tailored for diabetes prevention and self-management among NSB women should factor in patriarchy as an important social determinant of health.

## Introduction

### History and background of the Nepali-speaking Bhutanese (NSB) population in Pennsylvania

The Nepali-speaking Bhutanese (NSB) community is a rapidly growing population in Central Pennsylvania (PA). The NSB are Nepali migrants who were residing in Bhutan since the late 19th century up until the 1980's when Bhutan implemented a "One Nation, One People" policy to promote homogenous ethnic and religious national identity [1, 2]. As a result, the policy made adopting Bhutanese language, religion and dress code mandatory as well as removed Nepali language from school curricula. This policy came as a direct attack to the NSB population and was met with protests and resistance [3]. Protestors were seen as anti-nationalists and were arrested and detained; many were tortured, raped and even killed in detention. Approximately 100,000 NSB fled and sought refuge in neighboring countries, Nepal and India; the majority of whom ended up in refugee camps in Nepal, where they remained for 18 years [4–6]. The United Nations High Commissioner for Refugees (UNHCR) resettled the majority of the NSB population in the US, making them one of the largest refugee groups to resettle in the country [3].

After multiple resettlements, adapting to the US healthcare system with limited health literacy has been a challenge for the NSB population. Moreover, they face structural barriers such as lack of English proficiency, lack of transportation and lack of proper income [7]. They also encounter unique health challenges due to traumatic experiences of fleeing their homes, adverse living and health conditions in the refugee camps, and the psychosocial stressors that come with acculturation in third country resettlement [8].

In the Harrisburg area of PA, Hershey Medical Center (HMC), a tertiary health care center with Penn State Health, recognized a sudden influx of NSB patients in 2016. HMC identified the need to assist this growing population's transition into the healthcare system and began a dedicated NSB clinic. The clinic was established to intentionally provide care in a more respectful and culturally appropriate manner.

Anecdotal evidence from the clinic suggested an alarmingly high prevalence and poor management of type 2 diabetes. South Asian immigrants within western countries have been documented to have higher rates of diabetes and cardiovascular diseases than their white counterparts [8]. The NSB population share a similar risk profile for cardio-metabolic disorders with other South Asian immigrants [8]. Observations in the NSB refugee community in Northeast Ohio indicated excessive weight, particularly among women and poor dietary

practices [9]. Through the NSB clinic operated by the HMC, a clear need was identified for a community-based diabetes education program focusing on improving communication, addressing prevailing misconceptions, and culture-specific strategies.

## Community-based diabetes education program tailored to the NSB population

In 2019, a community-based diabetes education program was piloted in Harrisburg, PA which consisted of diabetes education and cooking demonstration workshops.

Observations from the workshops showed that although male and female patients were invited in equal numbers, the majority of the participants who attended the workshops were male. Yet, participants reported that the primary household cooks and caretakers were women. This could be an indication of the status of women in the NSB community, their healthcare access, autonomy, and ability to manage their diabetes.

Gender roles in the NSB community are clearly demarcated. Women lack the ability to make decisions regarding household and financial issues as well as equal access to the family's resources [10]. Women's access to education is constrained by traditional female roles such as domestic chores, child bearing and rearing. Evidence also suggests a high prevalence of gender-based violence among this population [11]. The above mentioned expressions of patriarchy are further exacerbated while seeking refuge and migrating to other countries [12]. Refugee experiences involve more hardships for women that consist of discrimination, violence, and trauma [13].

Despite growing numbers and rigid patriarchy in their culture of origin, the health status of NSB women remains under-researched; specifically, from a feminist perspective examining the experiences of NSB women at the intersection of gender, ethnicity, culture, and residency status. Hence, this study aims to understand the manifestations of patriarchy and its impact on NSB women's diabetes self- management.

## Conceptual framework

For the purpose of this study, Walby's 1989 definition of patriarchy was applied. In her book "Theorizing Patriarchy", she defines patriarchy as a system of social structures and practices in which men dominate, oppress and exploit women; where social structure rejects both biological determinism and the notion that every individual man is in a dominant position and every individual woman in a subordinate one [14]. This definition views patriarchy as a structural phenomenon rather than one perpetuated by the individual exploitative man [15]. She identifies a model of six structures of patriarchy; the patriarchal model of production, patriarchal relations in paid work, patriarchal relations in the state, male violence, patriarchal relations in sexuality and patriarchal relations in cultural institutions [14]. These structures constitute different forms of patriarchy that are present in society and can be interrelated or autonomous [15].

Although all six structures of patriarchy can directly or indirectly be linked to NSB women's diabetes self-management, this study will only focus on the three that are most closely linked i.e. culture, state and household production. Walby defines culture as one of the structures of patriarchy that composes the cultural notions of gender and the representation of women within a patriarchal gaze [16]. The state is defined as a set of social institutions which a monopoly over legitimate coercion has given territory. This structure consists of gendered political forces and the legal and political system that do or do not exist to protect women from patriarchal power in society [16]. Walby discusses household production with regard to gender relations in households including women's choice in marriage, their role in reproduction and

child rearing, their decision making rights within the households and the division of domestic labor among male and female household members [14]. Cultural components such as religion and gender norms, multiple resettlements due to statelessness and unpaid household production are hypothesized to impact NSB women's diabetes management the most among Walby's structures of patriarchy.

The structures of patriarchy are expected to impact the predictors of health such as healthcare access and autonomy, access to a healthy lifestyle, self-esteem and self-efficacy.

Healthcare access includes gaining entry into the healthcare system, access to a location where the needed healthcare services are offered, and finding a provider you can trust and communicate with [17]. Health autonomy is an individual's ability to make independent decisions without the coercion of others to make an informed decision regarding their health [18]. Healthy lifestyle is a way of living that helps to keep and improve people's health and well-being [19]. Additionally, self-esteem is an individual's subjective evaluation of their own worth [20]. Low self-esteem leads to mental health issues and poor overall health behaviors such as food consumption, exercise and substance use [21]. Hence, self-esteem is related to an individual's self-efficacy, which is an individual's personal belief of their abilities to execute courses of action required to deal with potential conditions [22]. In turn, these predictors impact the main outcome of the study, i.e. Type 2 diabetes (T2DM) self-management among NSB women. T2DM is a chronic, metabolic disease that occurs due to elevated blood glucose levels [23]. Over time, if left unmanaged, it can lead to serious damage to the heart, blood vessels, eyes, kidneys and nerves [23]. Diabetes mellitus is a chronic disease that can affect all aspects of life. Much of the care plan for this disease is interwoven with the daily life behaviors, thus diabetic individuals are the most responsible for control and management of the disease [24]. Diabetes self-management behaviors include physical activity, healthy eating, medication taking, monitoring blood glucose, diabetes self-care related problem solving, reducing risks of acute and chronic complications, and psychosocial aspects of living with diabetes [24].

To optimize diabetes management, patients living with diabetes should have regular screening tests and carefully monitor their blood glucose level [25]. It is also important for patients to have the skills and knowledge to effectively self-manage their condition. This can be a challenge for ethnic minority communities who do not have equal access to healthcare services due to lack of insurance, poor health literacy, language barriers, existence of deep rooted cultural and religious beliefs and lack of acculturation [25]. These factors also affect the diagnosis of diabetes in a timely manner [25].

In communities that are not individualistic, healthcare autonomy and autonomy support from family and friends are important [26]. A study found that perceived autonomy support from patients' informal health supporters such as family members and friends was associated with lower diabetes distress, greater diabetes management self-efficacy, and better self-monitoring of blood glucose [26].

Research has proven that adopting a healthy lifestyle contributes toward diabetes prevention and management. A large-scale German study found a strong association between the reduction in relative risk of developing major chronic diseases, such as diabetes, with healthy lifestyle factors such as smoking status, body mass index, physical activity, and diet [27]. Another study conducted among Asian Indians found that lifestyle modifications and use of metformin significantly reduced the incidence of diabetes in the Asian Indian population [28]. These findings were consistent with studies conducted in Japan and Finland [29, 30].

Additionally, other studies show that self-efficacy plays a prominent role in diabetes self-management and its outcome. An Iranian study showed that self-efficacy was the strongest predictor of diabetes self-management behavior [31]. When positive attitudes towards self-management increased, adherence to recommended diabetes self-care practices increased as

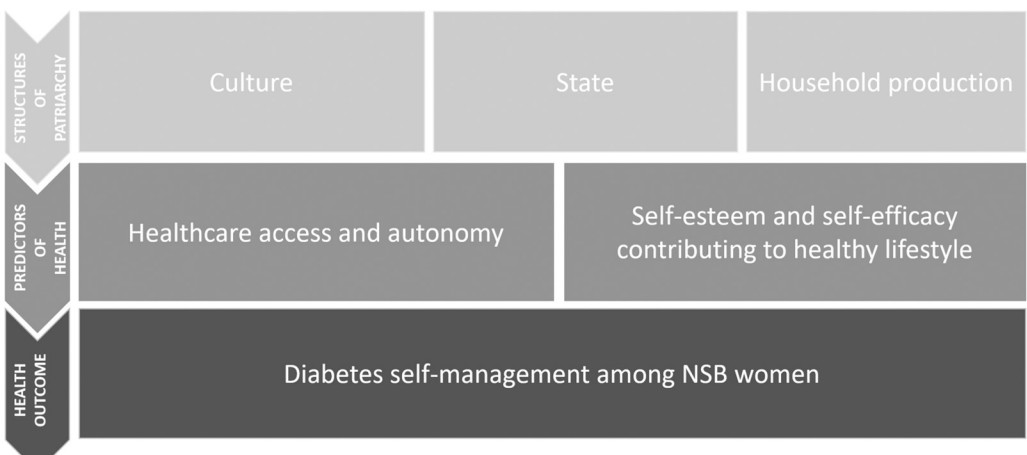

**Fig 1. Conceptual framework.**

well and led to controlled diabetes [30]. Another publication conducted on diabetes self-management behavior found a significant relationship between self-efficacy and quality of life (QOL) domains such as physical, psychological, social relations and environmental domains [32]. Self-efficacy was also associated with diet, exercise, self-monitoring blood glucose and foot care [33].

In Walby's structures of patriarchy, the predictors and outcomes of these studies were combined to develop a conceptual framework that guided this research. Fig 1 provides a visual presentation of the conceptual framework.

## Methods

### Research design

The research design for this study is an exploratory feminist qualitative inquiry. Exploratory qualitative inquiry allows for investigation of participants' lived experiences, subjective opinions and perceptions that provides the researchers a comprehensive understanding on the subject required for analysis and interpretation of the experience [34–37]. This study follows three main principles of feminist research: purpose, researcher-respondent, and reflexivity [38, 39]. Purpose emphasizes voice from marginalized populations and their contributions; researcher-respondent breaks down the hierarchy between researcher and participant balancing the power between the two; finally, reflexivity requires the researcher to center participants' voice to maximize objectivity [38, 39].

This inquiry was guided by the following central research question: How does patriarchy influence diabetes self-management about Nepali-speaking Bhutanese women living in Harrisburg, PA? The research question was guided by ideas, perspectives and assumptions that the researcher gained while previously working with the NSB population on their diabetes management.

### Recruitment criteria

Eligibility criteria for participation included the following: female participants from the NSB community; participants with T2DM; residing in Central PA; and ages 20 to 75 years old.

## Recruitment method

Fifteen participants were recruited from Hershey Medical Center via referral by clinical physicians from the Department of Family and Community Medicine. Physicians determined eligibility of participants by comparing their electronic medical records to the inclusion criteria from August 2019 to October 2019. A total of 29 individuals were identified by physician study partners and were deemed eligible for the study. All 29 were contacted by the principal investigator to explain the purpose of the study and recruit. Fourteen individuals did not respond or declined to take part in the study, leaving a total sample of 15. After 15 interviews, recruitment was terminated because investigators determined that data saturation had been reached and no new information emerged. Those who were interested in participating in the study were contacted by the first author, a native Nepali speaker, to provide further details about the study. Interested participants provided verbal consent to participate in the study. Each participant was provided with $30 gift card for their time and participation. A summary of participant recruitment is shown in Fig 2.

## Procedure

Semi-structured virtual interviews, field notes and observation were used as a primary data collection technique. Semi-structured interviews allow interviewers to follow the structured portion as a guide to keep them focused but also ask follow-up questions based on responses [39, 40].

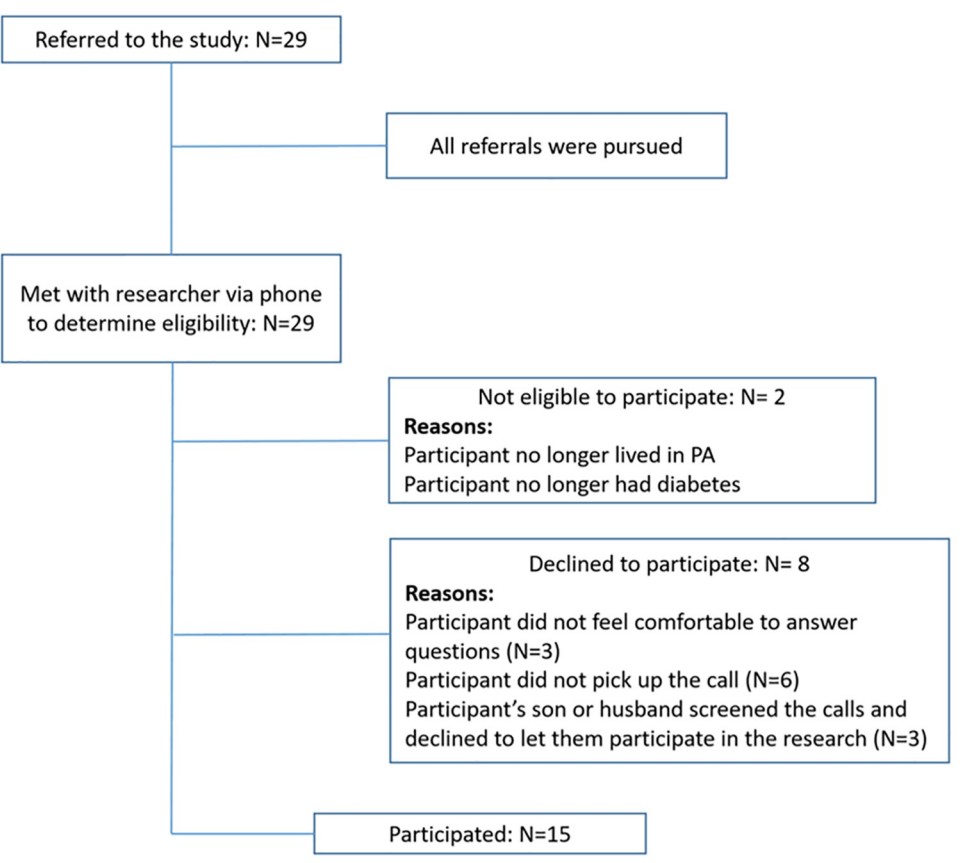

**Fig 2. Summarizing the referral and recruitment details of the study.**

Interviews were conducted between August-October, 2020. Semi-structured interviews were conducted over the phone using the Doximity phone app reflecting an HMC caller ID phone number. Although the Doximity app allows for video calls, we opted for audio calls because video calls require substantial need of technology such access to smart phone/computer, proper internet connection and the ability to use those technologies.

The principal investigator, who is Nepali-speaking, translated the interview guide into the Nepali language. The interviews were conducted in Nepali language by the principal investigator on a date and time that was most convenient for the participants. Interviews followed an interview guide which was developed in collaboration with all authors (S1 File). One volunteer was selected based on their active participation in the community-based diabetes program to take part in a pilot interview following the topic guide. The experience and content of this interview was discussed. The volunteer provided feedback and conformation that the guide was useful and appropriately tailored for the audience. Participants were first asked to self-report demographic information including their age, education level, marital status, household size, languages and religion. We asked participants questions about their diabetes management, family dynamics, gender roles, cultural norms, access and autonomy to healthcare. Examples of survey questions are: "Can you please describe a normal day's meal for me?", "Do you practice menstrual restrictions related to entering the kitchen and cooking in your household? If so, what are they?", "Do you participate in religious fasting? Why?", and "What do you know about your health insurance?". Each interview lasted between 20–45 minutes and was audio-recorded, with field notes made during each interview.

## Data coding and analysis

Interviews were transcribed verbatim in Nepali and translated into English by the principal investigator in preparation for analysis. Any names or other identifying information were redacted from transcripts throughout the transcription process. Field notes and observations related to each participant were included in the transcript as footnotes. The method of analysis was both data-driven and theory-driven. The codebook was developed inductively, based on interview transcripts to enable flexibility and detailed insight, as well as deductively to separate the data according to the theoretical framework. Two investigators independently read the interview transcripts and performed line by line open coding for key concepts that emerged from the data. NVivo version 12 was used to organize and analyze the data. Initial codes were discussed, compared and iteratively fine-tuned through consensus of the two data analysts. Complete descriptions of each code were generated, revised and agreed upon by both analysts. The codes were organized into several over-arching categories and further into themes as per the theoretical framework.

A thematic analysis was conducted as it has been identified as an approach well-suited to under-researched areas, enabling flexibility and detailed insight [41]. Braun and Clarke's six-step thematic analysis was conducted: becoming familiar with the data; generating initial codes; searching for themes; reviewing the themes; defining the themes; and writing-up the report [41].

After generating initial codes, the most prominent quotations were listed for each code within the codebook. The codes were then distributed into broader categories. Themes were developed inductively through those categories as well as through an a priori approach using the conceptual framework. For each theme all the included quotations in the codebook were synthesized to bring out the main ideas. Important quotations were presented in the findings along with the synthesized data. The over-arching themes were discussed with the research team to establish an acceptable thematic map of the data. The final codebook with the thematic map is shown in Table 1.

**Table 1. Participants' demographic characteristics.**

| Characteristics | Count | Percent (%) |
|---|---|---|
| **Age** | | |
| 30–39 | 2 | 13.3 |
| 40–49 | 4 | 26.7 |
| 50–59 | 6 | 40 |
| 60–69 | 2 | 13.3 |
| 70–75 | 1 | 6.7 |
| **Caste** | | |
| Brahmin/ Chhetri | 11 | 73.3 |
| Gurung | 3 | 20 |
| Tamang | 1 | 6.7 |
| **Religion** | | |
| Hindu | 13 | 86.7 |
| Buddhist | 2 | 13.3 |
| **Number of resettlements** | | |
| 3 | 11 | 73.3 |
| 4 | 4 | 26.7 |
| **Marital Status** | | |
| Married | 12 | 80 |
| Widowed | 1 | 6.7 |
| Divorced/separated | 2 | 13.3 |
| **Educational level** | | |
| Illiterate in Nepali and English | 10 | 66.7 |
| Primary | 1 | 6.7 |
| Secondary | 1 | 6.7 |
| University | 1 | 6.7 |
| Adult literacy classes | 2 | 13.3 |
| **Occupation** | | |
| Factory worker | 4 | 26.7 |
| Self-employed | 1 | 6.7 |
| Unemployed | 10 | 66.7 |
| **Household size** | | |
| 1 to 2 | 1 | 6.7 |
| 3 to 5 | 5 | 33.3 |
| 6 to 8 | 9 | 60 |

## Validity and reliability of analysis

Several quality criteria were used to ensure the validity of the results and interpretation.

1. Two qualitative data analysts with different ethnic background performed the analysis. This was to avoid any bias that could arise from the principal investigator (PI) sharing similar ethnic characteristics to the study participants.

2. To ensure the reliability of the coding and classification process, a coding comparison query that compares coding done by two experts in NVivo, was performed by calculating a Cohen's kappa coefficient. The kappa coefficient can range from -1 to +1 (+1 corresponding to a perfect concordance between the two experts). In the first instance, coders obtained a Cohen's kappa coefficient of 0.92 demonstrating excellent agreement [42].

3. Repeat interviews were carried out as a means of member checking. Participants were provided with an oral summary of their transcripts and asked if they wanted to change or remove anything they said [43]. Participants mostly agreed with the transcripts and minor revisions were made based on their review. The main themes that emerged from the transcripts were discussed with the participants to confirm that they identified with them [43].

4. Theoretical validation was conducted by comparing the results with existing scientific data (see Discussion).

## Ethical considerations

The study received ethical approval from the Institutional Review Board (IRB) at the Penn State College of Medicine. All the researchers involved in this study underwent ethics and data management training. Participants provided verbal consent. The "Consolidated criteria for reporting qualitative research checklist" was used to report the results [44].

Per IRB approval, the electronic data was stored in the Public Health Sciences Department's password protected file server with limited access only to study team members. Each transcript was given a unique ID to maintain anonymity and confidentiality before the data were analyzed. Securing physical space and privacy for participants during virtual interviews can be ethically challenging. The participants in this study were asked to go to a safe and private space. However, the researchers were unable to control participant's environment to ensure total confidentiality.

## Results

### Participants' demographic characteristics

Fifteen NSB women with diabetes were included in this study. The mean age was 50 years old. A vast majority were from Brahmin or Chhetri caste (n = 11), and others were from Gurung (n = 3) and Tamang caste (n = 1). Brahmins are at the top of the Hindu caste system in Nepal, followed by Chhetris. Gurungs and Tamangs are part of the non-Hindu indigenous nationalities also termed as *Adivasi/Janajati* [45]. Thirteen women followed Hinduism and two followed Buddhism as their religion. All women had been resettled three or more times. Twelve of the fifteen were married, 2 were divorced or separated and 1 was widowed. The majority of the women (n = 10) were illiterate in both Nepali and English. Two women had been to primary and secondary school respectively, two women attended adult literacy classes to acquire basic skills like reading, writing, math and English-language proficiency and one had a university degree. Only five women were employed, four of whom were employed as warehouse packagers and one of them worked in their family-owned restaurant. Nine women lived in large multigenerational households with 6 to 8 people, while five lived in households with 3 to 5 people, and only one lived in a two-person household. The participants' demographic characteristics are presented in Table 1.

### Themes

Six themes were formed from the data: 1) Cultural influences such as family structure, religious beliefs, traditional healthcare and gender roles determined NSB women's patriarchal upbringing and lifestyle; 2) Unpaid household production was largely influenced by patriarchy and was dependent on women; 3) Multiple (forced) immigrations have led to poor socioeconomic indicators and marginalization of NSB women; 4) Women's access to healthcare (including

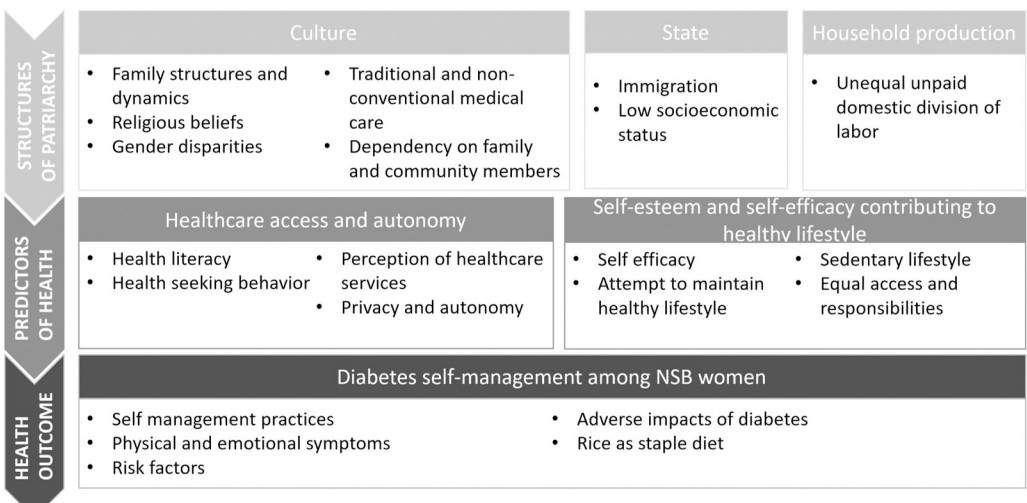

**Fig 3. Themes and categories represented within the conceptual framework.**

diabetes) was largely dependent on other family members due to poor financial, healthcare and overall autonomy; 5) Women's ability and attempts to maintain a healthy diabetes lifestyle was determined by their physical health condition, knowledge regarding good dietary practices and confidence to manage their disease and; 6) Women experienced adverse physical and emotional symptoms related to diabetes amidst their attempted adherence to diabetes self-management. The themes are represented within the conceptual framework in Fig 3.

The themes and supporting data are described below. The frequency of the references to key codes are presented in S1 Table.

**Theme 1: Cultural influences such as family structure, religious beliefs, traditional healthcare and gender roles determined NSB women's patriarchal upbringing and lifestyle.** *Family structure.* NSB women lived in large multigenerational households, mostly ranging from 6 to 8 household members. One woman described being raised in a polygamous household and another was in a polygamous marriage herself.

A majority of NSB women were married at a young age. Twelve out of fifteen women who participated in this study were married before the age of 18, among whom 4 were married before the age of 15.

"*I was really young. I was 12. My husband was 14. We married in Bhutan.*"

Although some women chose their marriage partners themselves, most marriages were arranged by their families and women claimed that they did not "*have any say*" in choosing their life partner. One woman recalled,

"*My parents and his parents decided. I did not have a say. I was really young.*"

Another reiterated,

"*Of course, it was arranged (laughs). There was no concept of love or love marriage at that time. I had never even heard of it.*"

*Religious beliefs.* The most common religion followed by NSB women was Hinduism, followed by Buddhism. Women who followed Hinduism practiced religious fasts as well as menstrual restrictions. Some women who had given up religious fasts after their diabetes diagnosis expressed feeling guilty about being unable to take fasts and said they "*asked God for forgiveness*". Women who continued to fast despite their diabetes diagnosis associated fasting with their husbands' long life and thought of it as good karma. Many women also stated that they were not sure about why they practiced such fasts.

> "*Yes. I take a lot of them [fasts]. During Ekadashi, Teej, Panchami etc. . ..Teej is for husband's long life. But I am not sure about all the other ones. Everyone takes it. Maybe something good will happen. It is good karma.*"

All women, except one, disclosed that they practiced some forms of menstrual restrictions due to religious beliefs. They said they were not allowed to enter the kitchen or cook for themselves or their family during their menstruation. One particular woman described her reasons;

> "*My husband cannot tolerate it. . ..It's not that he cannot tolerate but he gets allergies and other physical reactions if I cook when I'm menstruating.*"

However, most women did not know the reason behind the practice. They displayed a certain conditioning to accept their culture without question.

> "*It [menstrual restrictions] is part of our culture. Just because we are in the USA now doesn't mean we forget our culture. It has been going on for generations. It is also a time when I get rest. This is a culture that I am going to continue following and I cannot change it.*"

*Traditional and non-conventional healthcare.* Many participants visited traditional healers (i.e. *dhaami)* for some of their health issues, apart from their regular doctor's visit. They did not take any medications prescribed by the traditional healers but mentioned that it was important for them to see *dhaamis* as part of the healing process.

> "*There are not health issues right now so I don't go but when we are sick we need dhaami. When we were in Bhutan we used to go often.*"

Many also used Ayurvedic or homeopathic medication for some of their health issues.

> "*I take a lot of Ayurveda medicines. My son and daughters-in-law order it online for me. My pain got better only because of that, I feel.*"

*Gender disparities.* Most of the participants in this study were illiterate. However, many mentioned that their husbands or their brothers had access to education. One woman stated,

> "*My brothers are quite educated. My oldest brother can read and write but other brothers have pursued higher education.*"

Another mentioned that her household responsibilities did not leave time for her to pursue education.

"*My husband. . . .was a teacher. He died only after 7 years of marriage. I had already had three children then. He was a good man. . . .He married me with the promise to educate me. His family was very big and I never got the time to study from doing all the household chores.*"

When women were asked to define a good woman in their community, many of them said they did not know how answer that question. Others defined a good woman as soft-spoken, well-behaved, nurturing, and one even mentioned being educated.

"*I think a good woman is who is well behaved- who gets along well with their family and someone who helps other people.*"

*Dependency on family and community members*. Women spoke about receiving family support to self-manage their diabetes. Many of their family members were employed as their caretaker, under the PA waiver program. However, a tendency to depend on others was also observed. Most of the women did not drive themselves but claimed that they were not inconvenienced by it as they asked other family members or community members.

"*She's [daughter] learning [to drive] right now. Meanwhile, some relatives and other community members take us.*"

**Theme 2: Unpaid household production was largely influenced by patriarchy and was dependent on women.** Women who were unemployed were highly dependent on their female family members such as daughters-in-law and daughters for support. The daughters-in-law and daughters, despite having full-time jobs or being full-time students, held most of the household responsibilities of such as doing the laundry, grocery shopping and taking care of the children and the elderly in the family.

"*My daughter-in-law does all the laundry, cooking. She does everything. I feel very dizzy and I get a headache. That's why I don't do any work.*"

The men in house were not taught or expected to do any household chores.

"*My brother was never expected to do household chores ever since our childhood so he is not used to it. He does some grocery shopping for us. My sister-in-law does most of it.*"

The daughters-in-law and daughters were also the primary cooks in the family. The women explained that men only went into the kitchen for cooking when the women in the house were menstruating and were forbidden to enter the kitchen. When asked if the men in their family knew how to cook, one woman stated;

"*No, he does not. My daughter has to teach him from outside the kitchen (laughs).*"

In one case, the woman said that although all the household responsibilities were hers, she believed that they should be equal.

"*Most of the household chores are my responsibilities. Only if I tell my husband, this work is too much for me, then he will help a little bit. Otherwise, it's all on me. He only does the shopping because I cannot drive. I get angry when he doesn't help in the house. I believe we should do the housework equally.*"

By contrast, five out of fifteen women mentioned that they shared household responsibilities with every member of the household.

**Theme 3: Multiple (forced) immigrations have led to poor socioeconomic indicators and marginalization of NSB women.** *Immigration*. The NSB women had been through multiple resettlements due to their former refugee status. They were all born in Bhutan and had lived in refugee camps in Nepal. They were resettled to the US. Some of them had lived in other states in the US before settling in Harrisburg. The women described their life in Bhutan as pleasant and simple as farmers. They recalled,

> "*We used to do agriculture work in Bhutan. In Nepal we were not allowed to work. I like it here in the USA. The access to health is better here. I got sick in Nepal a lot but I had to stand in line there all the time. In Bhutan, I only got vaccinated but I delivered all my children at home.*"

While some saw moving to the US as an opportunity to start a new life, few women were displeased with the resettlement.

> "*I don't know what to say. If I say, I don't like it, I don't think they will take me back to Nepal. Half my family is Nepal. I want to be here and there.*"

*Low socioeconomic indicators*. Majority of the NSB women had received little to no formal education due to lack of schools in Bhutan. Many specified lack of family support for their illiteracy.

> "*I can't even read Nepali. I never went to school. None of the girls went to school when I was in Bhutan. I guess my parents didn't want us to go to school. I don't know.*"

Due to their lack of access to education, most NSB women mostly only knew how to speak Nepali but could not read or write it. They did not understand English. Living the US, their lack of language proficiency also limited their mobility.

> "*I don't know anything. I don't go out of the house very often. I only speak Nepali.*"

This impacted their access to healthcare as they were not able to call the healthcare institutions themselves to make appointments or receive calls from the hospital.

> "*I never go to the hospital alone. My brother or my sister always come with me. They have to translate for me.*"

Only five of the fifteen NSB women were employed. While one of them worked in the kitchen of her family owned restaurant, others worked in factory warehouses doing packaging jobs. Both jobs required minimum communication and interaction with others and was suitable for them due to their lack of English speaking abilities.

> "*I work in a warehouse. Uneducated people like us only get jobs in such places.*"

Only one woman in this study knew how to drive. Others did not know how to drive or had access to their own mode of transportation. Women who were employed relied on other members of their family or community members to drive them to work.

"*When I first came to USA, I thought without learning how to drive it will be difficult to survive. I tried a lot to learn but it was too hard for me. My son discouraged me because he got scared I might get into an accident.*"

**Theme 4: Women's access to healthcare (including diabetes) was largely dependent on other family members due to poor health literacy and poor financial, healthcare and overall autonomy.** *Health literacy*. NSB women displayed poor health literacy such as poor understanding of diabetes, poor knowledge of health insurance and skepticism of western medicine. Almost all women lacked awareness and proper knowledge on diabetes even though they insisted that their doctors explained it to them.

"*I feel good one day, I feel sick the next. It goes high and low. I don't understand at all how it works.*"

As the participants were all recruited from a medical center, they were all insured. However, most women did not know anything about their insurance status and whether or not they were insured.

Upon diagnosis, some women were skeptical of taking medication, scared it would cause dependency. One particular woman did not believe the lab results as she did not feel unwell.

"*They just said you have diabetes and high blood pressure, you need to take medicine. But I have no symptoms so I didn't care so much.*"

*Healthcare seeking behavior*. Most women had a reactive approach to healthcare, did engage in preventive medical care, and only sought medical help when they developed symptoms. All but one were diagnosed with their diabetes during a regular doctor's visit rather than due to their symptoms. Many of them were diagnosed during their immigration process when they had to get mandatory health screenings, either in Nepal or the US.

""*Yes, we had to get all our medical examinations during our immigration process before coming to America, so took my blood and when the report came, that's when they told me I have diabetes.*"

However, after arriving in the US, most women went for regular doctor's visits for their diabetes. Due to COVID-19, many women stated that they missed appointments with their doctors and diabetes educators. Most women preferred to use their family members as interpreters during their medical appointments, while some used the medical interpreters provided by the healthcare center.

"*My son and daughter in law translate for me mostly.*"

*Perception of healthcare services*. The women mostly had a positive attitude towards their healthcare provider. Most women preferred having a same sex provider.

"*I can talk with a male provider too but I am more comfortable to talk to a female provider.*"

*Privacy*. The interviews for this study were conducted virtually and the interviewer was not able to control the level of privacy for the interviews. While calling the participants, the

women's husband or son would first pick up the call, in many cases this appeared to the author like gatekeeping. Some male family members asked in detail about what the call was about before passing the phone to potential participants, asked to call back and declined on their behalf to participate in this study. Many others hovered in the background during the interviews, even while repeatedly being asked to leave the room for privacy. In many interviews, the male family member was also heard providing answers for the women.

*Autonomy*. Most NSB women lack financial autonomy. Older women mostly relied on their sons and daughters-in-law to make financial decisions. Younger women, even those who were employed, relied on their husband to make such decisions. Some women said they were not able to identify currency as they were illiterate and innumerate.

"*I don't even know how to read the numbers in the notes. I have never dealt with money. If I have a note, someone has to tell me how much that is. I never need money though so I never have to use it.*"

In terms of healthcare autonomy, women lacked decision making ability to go see their healthcare provider for ailments. When asked how they make the decision to see their provider, most women answered that it was a family decision and someone had to take them to the appointments.

"*Everyone does it [makes healthcare decision for me]. My daughter-in-law does everything. Would you like to speak to her instead?*"

Women were not comfortable communicating with their healthcare provider due to the language barrier. Moreover, they were also not aware of their healthcare rights.

"*I don't think we have a choice [to choose a same sex provider] in America. You have to take whatever they give us.*"

**Theme 5: Women's ability and attempts to maintain a healthy diabetes lifestyle was determined by their physical health condition, knowledge regarding good dietary practices and confidence to manage their disease.** *Attempt to maintain healthy lifestyle*. Despite their lack of knowledge on diabetes, most women discussed their attempts at portion control and restricting their diet as a measure to control their diabetes.

"*I only eat one big meal with rice for lunch. Otherwise I eat small portions throughout the day.*"

Some women were aware of physical activity as an essential part of diabetes management. Although many did not exercise regularly, they counted their physically demanding work and house chores as physical activity.

"*My work is physically demanding so that it is my exercise. I get up in the morning and I cook and clean the house so I also get my exercise from there. Maybe I need to start by doing little exercises.*"

*Sedentary lifestyle*. Most unemployed women had limited mobility because of their illnesses and had a sedentary lifestyle. They had other female family members to do their chores and take care of them.

"*I don't do much, I go around the house. I stay indoors mostly. Sometimes I get bad backache so I just lie down in my bed.*"

*Poor self-efficacy.* Almost all participants claimed to be motivated to manage their diabetes. They essentially wanted to 'not be sick anymore.'

"*I do all the exercises so my diabetes doesn't go high. I am motivated. I control my diet. You understand how Nepali diet works. I like to eat corn a lot and it's the season for it so I am tempted to eat that. But I try to control most times.*"

However, only few of them were confident that they were successful in managing their diabetes. Others were eager to 'get rid' of this disease but were not sure how to do that.

"*I try but I cannot understand diabetes at all. I don't understand when it goes high or low.*"

**Theme 6: Women experienced adverse physical and emotional symptoms related to diabetes amidst their attempted adherence to diabetes self-management.** *Risk factors.* Most women reported a history of diabetes in their family. Three out of fifteen women reported that their diabetes started during their pregnancy and never went away. Almost all women reported having multiple comorbidities, most commonly hypertension and high cholesterol.

"*I have thyroid, cholesterol, high blood pressure. They say a lot of things, I can't even say the name of most of the diseases, Madam.*"

*Cultural diet as challenge to maintain diabetes.* Rice was an important part of women's cultural diet. Participants listed restricting rice-based diet as the biggest challenge of living with diabetes. Almost all participants revealed replacing white rice with '*bagada*' rice or 'diabetic' rice, as they would call it. They perceived it to be a "*healthier and diabetes friendly*" alternative to plain white rice. While many women claimed to be trying to restrict their rice intake, one woman exclaimed;

"*We all have to die one day. I will die eating rice.*"

*Physical symptoms.* Some women experienced mild to moderate symptoms when their sugar levels were high or low such as headache, loss of appetite, fatigue and most commonly dizziness. Others experienced more severe symptoms such as losing consciousness, losing the ability to walk or talk, shivers and severe sweating. Women complained of chronic pain that prevented them from doing their daily activities.

"*I don't do any work. I have so much pain. I don't go to the kitchen at all. My breast feels like it is burning all the time. I just sit in my bed the whole day.*"

Three women also had to seek urgent medical assistance due to diabetes related complications.

*Emotional symptoms.* Although most women reported stress related to diabetes management, some women reported extreme fear of diabetes symptoms. These fears were based on misconceptions about the disease.

"*I think that because of my sugar and pressure [hypertension], I may fall and get paralyzed or even worse, die. That's what scares me.*"

Another cause of stress for women was their dependency on others, followed by stress related to family and work.

"*I take a lot of stress. I can't speak English, I can't drive. I have [a] child. I want his father to come to the USA. I always have to wait for other people to go to the health care facility and sometimes when no one is available it's really very stressful for me. I am mostly stressed about my heart disease and now more diseases are adding up. I have to work a lot in the USA. We have lots of overtime right now we can't even sleep properly. My father recently passed away and I noticed that my diabetes started after that.*"

*Adverse impacts of diabetes.* The biggest challenges of living with diabetes was their frustration managing their diabetes because of poor understanding of diabetes. Women who were unemployed blamed their illness for their limited mobility and their inability to work.

"*No, I don't have any work. I used to work around the house in Nepal. But since I got diagnosed with diabetes, I can't work at all. Since 2013, I can't do a lot.*"

## Discussion

Patriarchy was noted to affect NSB women's diabetes management in numerous domains.

### Child marriage and education

Arranged child marriage, a critical component of the patriarchal religious construct, was common among NSB women from all age groups in this study. Women lacked autonomy to decide the timing of and partner for marriage. Given the patriarchal nature of Nepali culture, daughters are often considered economic burdens to their families and are married off early to avoid financial costs of education and rearing [46, 47]. Although child marriage is globally recognized as a harmful practice, it continues in many developing countries due to social and religious customs. Moreover, young girls do not have the autonomy or the negotiation skills to go against their family's decisions [48]. The 2016 Nepal Demographic Health Survey (NDHS) reported that the median age at first marriage among women was 17.9 years [49]. While 13% of women were married by age 15 and 52% of women were married by age 18, only 3% of men were married by age 15 and 19% before age 18 [49]. Child marriage acts as a huge barrier for girls from entering or completing their education, which is a powerful tool for empowering women. Educated women and girls are more likely to be healthier, generate more income through employment in formal labor markets and ensure better healthcare for their family [50]. Furthermore, child marriage could be a factor for the majority of the study participants' due to its impact on low literacy levels, minimum wage jobs, or unemployment.

### Religious influences

The majority of the women in this study followed Hinduism, which is the predominant religion in Nepal. They practiced religious fasting based on Hindu beliefs. Religious fasting or *barta* is aimed at securing marital, financial, medical and social success and is mostly carried out by women and under women's guidance [51]. It is an indicator of the roles that women play within their families [51]. Although fasting is an important religious practice for Hindus,

fasting for long periods of time can be harmful for people with diabetes [52]. As a diabetic, it is important to eat regularly to ensure regulated blood sugar levels. A study on dietary behaviors among Nepali diabetics found that Hindu Nepali women preferred practicing religious fasting and felt guilt related to stopping because of their diabetes [53]. Women in this study also practiced various restrictions during their menstruation, such as entering the kitchen and cooking. Menstrual restrictions can act as a barrier for women to access a diabetes friendly diet during their menstruation period as they may lack agency to advocate for their dietary needs [53]. The laws embedded in Hindu norms are highly patriarchal in nature and place women in a subordinate position to men where their most important role is to bear children and take care of the family and house [54].

Women in the current study sought traditional healers in conjunction with their allopathic medicine. Traditional healers and Ayurvedic medicine is central to South Asian culture, although they haven't been proven as effective treatment for diabetes. These kind of health seeking behaviors could potentially cause delay in accessing medical healthcare services and act as a barrier to managing diabetes [55]. Women in this study displayed lack of healthcare autonomy. Hence, seeking traditional healers may not have been their personal decision. Moreover, they may have sought traditional healers as they are widely accepted in Nepali society and are more accessible due to similar cultural backgrounds and languages.

## Gender roles

In this study, culturally constructed gender roles largely influenced domestic division of labor and unpaid household production. Women, mostly daughters-in-law, were the primary caretakers and cooks in their household, sometimes in addition to their paid employment outside their homes. Similar findings were reported in several other studies conducted in Nepal [56, 57]. The NSB women in this study reported that male family members only cooked when the female members were menstruating.

Uprety's field report on gender and nutrition in Nepal reported that daughters-in-law had to cook for the entire family but were only allowed to eat once every member had been served or after the men had finished eating [58]. This could result in unequal dietary intake among men and women as reported by Sudo et al. as women may be left with unhealthier food options [59]. Although, women as primary cooks play an important role in influencing household dietary behaviors, many do not get to set the food menu. Their role as caretakers with no agency can likely take both a physical and emotional toll on them [59, 60]. In the current study, women held the primary role of cooking for their families but lacked awareness on healthy diets and sometimes lacked autonomy to decide the menu for the family. Women in this study also reported extreme emotional symptoms likely due to the double burden of their disease along with their role as primary caretakers.

## Immigration

The NSB women in this study had been through multiple resettlements starting as refugees in Nepal in the early 90s. Immigrants encounter economic, systemic, informational, cultural, and linguistic barriers to accessing support and services [12]. Women experience these barriers in a greater magnitude due to their multiple caregiving responsibilities, which limit their opportunities to learn the language of the host country and access the services. Women in this study came from lower socio-economic backgrounds which led to their poor access to education and overall literacy, access to informational resources and learning a new language. The intersection of gender, ethnicity, culture, and immigration status increases the risk of experiencing adverse manifestations of patriarchy, particularly conflicts in spousal relationships [12]. These

barriers were further magnified for women in this study as they had faced more than three migrations.

NSB women in this study demonstrated poor understanding of diabetes and diabetes management. Studies conducted to assess knowledge, attitude and practice regarding diabetes in Nepal found similar results of low diabetes-related literacy that resulted in low adherence to diabetes self-care recommendations. Proper knowledge and awareness of diabetes ensures the patients' ability to make informed decisions about their care [61, 62]. This study demonstrated that NSB women's ability and attempt to maintain their diabetes was determined by their knowledge of good dietary practices and self-efficacy.

### Exercise and physical activity

Physical activity stemmed from domestic chores and work-related activities in this study. Similarly, Kadariya and Aro reported that walking as the means to travel and work related activities contributed to physical activities performed by Nepali diabetics [63]. As part of Nepali culture, people do not purposely engage in exercise activities to remain physically active; it is simply achieved through routine activities during daily life [63]. Furthermore, for NSB women, higher body mass index is culturally viewed as a sign of prosperity. Hence, there is less social pressure to lose weight [68]. Therefore, physical activities, such as exercise, among NSB women is uncommon. This is worse among immigrants due to logistical barriers. Due to their poor socio-economic status, many have resettled in neighborhoods that are not conducive to walking or movements that promote physical fitness due to lack of a safe public places [64].

### Self-esteem and self-efficacy

The NSB women in this study suffered from overall low self-esteem due to their lower economic and educational status. They had low confidence with managing their diabetes due to their poor understanding of the disease. Another Nepali study on factors associated with non-adherence of self-management practices reported similar results [61]. Individuals who demonstrate stronger motivation and self-efficacy to manage their diabetes are more likely to initiate and maintain diabetes self-management practices [61].

### Socio-economic influences

The majority of NSB women in this study lived in large multigenerational households. Parajuli et al. found that individuals living in extended families had poorer adherence levels than those living in nuclear families, likely due to high resources sharing among family members. The study also reported low adherence among people with poor knowledge on diabetes and people from lower economic class [64]. Likewise, this study found low adherence and poor knowledge among NSB women.

### Dietary practices

Unrelated to patriarchy, the biggest challenge for women to manage their diabetes was associated with their diet compliance, especially limiting rice intake. White rice is a staple Nepali diet and is commonly a part of every meal. The Nepali term 'khana' meaning meal was used interchangeably by NSB women to talk about rice. Studies conducted among this population have shown similar findings on the cultural importance of rice in Nepali diet and the challenge to replace or reduce its consumption [65, 66]. In the present study, almost all NSB women controlled their portion of rice instead of cutting it off completely from their diet. They also replaced their white rice with 'diabetic rice' or 'bagada rice' which has a slightly lower glycemic

index than white rice. Diabetic patients are warned against consuming large amounts of white rice due to its high glycemic index which causes high blood sugar levels [64].

## Emotional symptoms associated with diabetes

Women in this study reported extreme emotional distress caused by their diabetes. Several studies conducted in Nepal associating depression with diabetes reported that low educational status and poor understanding of the disease was associated with poor mental health outcomes among diabetics [67–70]. Unemployment and poor financial status was also linked with emotional distress due to the financial burden caused by diabetes [71]. Women in this study had all the risk factors for emotional distress due to their diabetes; they were not educated, lacked proper knowledge regarding diabetes and either employed at blue collar jobs or were unemployed.

The NSB women appeared to have resigned to their diagnosis of diabetes, much like other aspects of their life such as child marriage, religious practices, migration and employment. They not only lacked awareness about their disease but also self-efficacy to self-manage their diabetes. This is likely due to their lack of autonomy regarding their life and health because of the patriarchal norms that have dictated their survival throughout their lifecycle.

Healthcare providers often believe that the responsibility for self-care should be undertaken by the patients themselves, assuming that patients, especially women from patriarchal cultures, have the autonomy to do that or seek support for it [72]. Findings from studies conducted on chronic illness among immigrant women of color are also in line with results of this study. They reported poor self-management due to lack of autonomy and household duties as well as elevated levels of emotional distress and depression [72–75].

## Strength and limitations of the study

This is the first qualitative study that explores patriarchy as a challenge for diabetes self-management among women. This study was guided by a conceptual framework which was used to inform the research question and address the primary objective of the study. Although the study used a small sample size, data saturation was achieved. Rigorous procedures were implemented to ensure the validity of the data such as a having multiple data analysts, using the Cohen Kappa coefficient for assessing agreement and member checking.

This study was conducted in Harrisburg PA among NSB women and the results may not be generalizable to a wider population. However, commonalities may be found among NSB women living in other parts of the world and other South Asian populations.

Due to COVID-19 restrictions, the interviews were conducted virtually which could not ensure full privacy for the participants which has been addressed in the paper. Participants were compensated with gift cards for their participation, which may have influenced their willingness to participate in the study.

## Conclusion

Insights from this study illuminate how patriarchal norms influence NSB women's diabetes self-management. As within the conceptual framework, the study found that the structures of patriarchy i.e. culture, household production and state impacted predictors of NSB women's health such as healthcare access and autonomy, access to healthy lifestyle, and self-esteem and self-efficacy. These in turn, influenced NSB women's diabetes self-management. Cultural practices that start early on within women's lives, such as child marriage, religious restrictions in addition to women's access to education and their autonomy related to finance and mobility, impacted their access to healthcare, knowledge regarding their diabetes and self-efficacy. In

the case of the NSB women, these influences were magnified due to their multiple resettlements as former refugees. Future lifestyle interventions tailored for diabetes prevention and healthy diabetes self-management among NSB women should factor in gender roles and patriarchy as an important social determinant of health. These interventions should consider women's literacy levels, their access to assets and resources, gender norms regarding unpaid household production as well as their autonomy in decision making and mobility. Further studies can include a more in-depth analysis of NSB women's lived experiences within the culture of patriarchy pertaining to their migration and health.

Although women's unpaid household production constrain them to maintain full-time jobs, their work is not monetarily compensated, leaving them dependent on other family members. This results in lack of financial and healthcare autonomy. It is not enough to state aims of equitable access to healthcare in policies. Systemic disadvantages such as the influence of gender roles on women's access to healthcare and disease management should also be taken into consideration. The perspectives of women whom policies affect need to be included in all areas of women's healthcare policy making.

## Supporting information

**S1 File. Interview guide.**
(DOCX)

**S1 Table. Codes used in qualitative analysis related to themes.**
(DOCX)

## Acknowledgments

The authors would like to express their gratitude to the participants in this study for contributing their time and their experiences to this research.

## Author Contributions

**Conceptualization:** Aditi Sharma.

**Data curation:** Aditi Sharma.

**Formal analysis:** Aditi Sharma.

**Investigation:** Aditi Sharma.

**Methodology:** Aditi Sharma, Heather Stuckey, Megan Mendez-Miller, Yendelela Cuffee, Aubrey J. Juris, Jennifer S. McCall-Hosenfeld.

**Project administration:** Aditi Sharma.

**Resources:** Aditi Sharma.

**Software:** Aditi Sharma.

**Supervision:** Heather Stuckey, Megan Mendez-Miller, Yendelela Cuffee, Aubrey J. Juris, Jennifer S. McCall-Hosenfeld.

**Validation:** Aditi Sharma.

**Visualization:** Aditi Sharma.

**Writing – original draft:** Aditi Sharma.

**Writing – review & editing:** Aditi Sharma, Heather Stuckey, Megan Mendez-Miller, Yendelela Cuffee, Aubrey J. Juris, Jennifer S. McCall-Hosenfeld.

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
