## [Decision Letter · Decision Letter 0]

20 Sep 2021

PONE-D-21-19730The influence of patriarchy on Nepali-speaking Bhutanese women’s diabetes self-managementPLOS ONE

Dear Dr. Sharma,

Thank you for submitting your manuscript to PLOS ONE. After careful consideration, we feel that it has merit but does not fully meet PLOS ONE’s publication criteria as it currently stands. Therefore, we invite you to submit a revised version of the manuscript that addresses the points raised during the review process.

We look forward to receiving your revised manuscript.

Kind regards,

Shyam Sundar Budhathoki

Academic Editor

PLOS ONE

2. In the ethics statement in the Methods and online submission information, please clarify whether consent was written or verbal.  If verbal, please also specify: 1) whether the ethics committee approved the verbal consent procedure, 2) why written consent could not be obtained, and 3) how verbal consent was recorded. If the need for consent or parental consent was waived by the ethics committee, please include this information.

Additional Editor Comments (if provided):

Reviewers' comments:

Reviewer's Responses to Questions

**Comments to the Author**

1. Is the manuscript technically sound, and do the data support the conclusions?

Reviewer #1: Yes

Reviewer #2: Yes

2. Has the statistical analysis been performed appropriately and rigorously? 

Reviewer #1: Yes

Reviewer #2: Yes

3. Have the authors made all data underlying the findings in their manuscript fully available?

Reviewer #1: Yes

Reviewer #2: Yes

4. Is the manuscript presented in an intelligible fashion and written in standard English?

Reviewer #1: Yes

Reviewer #2: Yes

5. Review Comments to the Author

Reviewer #1: Title: The influence of patriarchy on Nepali-speaking Bhutanese women’s diabetes self management

General comments

This manuscript is based on primary qualitative data collected through virtual interviews. The goal of this analysis is to understand the manifestations of patriarchy and its impact on NSB women’s diabetes self-management employing a conceptual framework based on Walby’s structures of patriarchy. This manuscript covers the unique characteristics of population. It will contribute to scientific literature on the patriarchy and its influence on self-management of diabetes among women. However, it does require substantial revisions.

Abstract

No comments

Introduction

It would be beneficial to consider the diabetes self-management status of migrant women in USA and other developed countries. Also, mention other scientific studies from similar cultures or where patriarchal society is prevalent to understand how patriarchal society impacts women in diabetes self-management.

Method

Recruitment: Please mention the rationale of contacting 29 people. Is it due to only 29 people are registered in the Hershey Medical Center and meet eligibility criteria or it is due to for convenience?

Line 179-180: Do clinical physicians contacted 29 people to recruit initially? It needs to be clear.

Procedure: Please mention that the Deximity app allows video call to clarify that observation was possible in virtual interview. Please use the one term for indepth virtual interviews. It appears in different names- interviews, semi-structured interviews.

Interview guidelines: Mention the process of translation of interview guidelines. Based on interview guide it appears 35 questions and mostly open-ended questions. However, in Line 200-208, all questions are not addressed as mentioned in the interview guide. If authors only highlighted main questions in the text, please mention it for clarity.

Ethical consideration: Please mention the ethical consideration related to data management and analysis. Also, please consider how issues with privacy related to virtual interview were addressed.

Result

The result is comprehensive however all the statements made should be substantiated by data. Some of the sub-themes do not have any data. Please include this. Also, since the age group of participants is diverse, is there any difference cited by younger and older participants in any themes and sub-themes?

Discussion

Overall, it has dense information. However, it is hard to follow since it has several paragraphs without headings. It would be better to mention themes as headings and discuss under them.

The authors have discussed well on the marriage and impact on education in a patriarchal society. However, this is missing information on how child marriage and education has influenced migrant women to self-management of diabetes. Please include this information to add richness to the finding of this study.

Paragraph 6: Please link how economic, systemic, informational, cultural, and linguistic barriers impact migrant women to manage diabetes.

Paragraph 8: Please mention scientific evidence to support ‘As part of Nepali culture, people do not purposely engage in exercise activities to remain physically active; it is simply achieved through routine activities during daily life’.

Paragraph 12: Please mention what are these several studies, “Several studies conducted in Nepal associating depression with diabetes reported that low educational status and poor understanding of the disease was associated with poor mental health outcomes among diabetics”.

Thank you for the opportunity to review your work.

Reviewer #2: This is a well written manuscript that provides a qualitative analysis of a small sample of Bhutanese women. I would have expected the manuscript to improve on two fronts before publication. First, refers the link with studies on patriarchy and overweight, as this is the main connection of the story. Second, provide slightly more practical implication in the discussion part, what are the policy implications of the results?

6. PLOS authors have the option to publish the peer review history of their article (what does this mean?). If published, this will include your full peer review and any attached files.

Reviewer #1: **Yes: **Mandira Adhikari

Reviewer #2: No

---

## [Author Response · Author response to Decision Letter 0]

6 Jan 2022

Dear Dr. Budhathoki, 

Thank you for giving us the opportunity to submit a revised draft of our manuscript titled 

“The influence of patriarchy on Nepali-speaking Bhutanese women’s diabetes self-management” to PLOS One. We appreciate the time and effort that you and the reviewers have dedicated to providing your valuable feedback on our manuscript. We are grateful to the reviewers for their insightful comments on our paper. We have been able to incorporate changes to reflect most of the suggestions provided by the reviewers. We have highlighted the changes within the manuscript. 

Here is a point-by-point response to the reviewers’ comments and concerns. 

Comments from Reviewer 1

Comment 1: 

It would be beneficial to consider the diabetes self-management status of migrant women in USA and other developed countries. Also, mention other scientific studies from similar cultures or where patriarchal society is prevalent to understand how patriarchal society impacts women in diabetes self-management.

Response: Thank you for pointing this out. We added a paragraph in the discussion section drawing similarities to our findings from other studies conducted on chronic illness among immigrant women of color. 

Comment 2: 

Please mention the rationale of contacting 29 people. Is it due to only 29 people are registered in the Hershey Medical Center and meet eligibility criteria or it is due to for convenience? Do clinical physicians contacted 29 people to recruit initially? It needs to be clear.

Response: Thank you for your comment. We added the recruitment timeline in the manuscript to reflect the rationale of contacting 29 people. We also clarified that the participants were contact by the principal investigator. 

Comment 3: 

Please mention that the Doximity app allows video call to clarify that observation was possible in virtual interview.

Response: Thank you for your feedback. We added a sentence in the manuscript to clarify that observation was possible through video calls with the Doximity app. However, we opted for audio calls because video calls require substantial need of technology such access to smart phone/computer, proper internet connection and the ability to use those technologies. 

Comment 4: 

Please use the one term for in-depth virtual interviews. It appears in different names- interviews, semi-structured interviews.

Response: Thank you for bringing this to our attention. We changed all terms to “semi-structured” interviews in the manuscript. 

Comment 5: 

Mention the process of translation of interview guidelines.

Response: We have added a sentence in the manuscript that the interview guide was translated by the principal investigator into the Nepali language. 

Comment 6: 

Based on interview guide it appears 35 questions and mostly open-ended questions. However, in Line 200-208, all questions are not addressed as mentioned in the interview guide. If authors only highlighted main questions in the text, please mention it for clarity.

Response: Thank you. The interview questions mentioned in text are examples of some questions included in the interview guide. We have revised the text to clarify that.

Comment 7:

Please mention the ethical consideration related to data management and analysis. Also, please consider how issues with privacy related to virtual interview were addressed.

Response: Thank you for the comment. We have added ethical considerations regarding data management, analysis and virtual interview in the text. 

Comment 8:

The result is comprehensive however all the statements made should be substantiated by data. Some of the sub-themes do not have any data. Please include this.

Response: Thank you for point this out. We have added some quotes as a form of data in the result section. 

Comment 9:

Since the age group of participants is diverse, is there any difference cited by younger and older participants in any themes and sub-themes?

Response: We appreciate your comment. The objective of the study was not to compare diabetes management across age. This would not have been possible due to the small sample size. We included different age groups to bring in diverse responses. However, this is an excellent suggestion and perhaps can be used for future, larger studies.

Comment 10: 

Overall, it has dense information. However, it is hard to follow since it has several paragraphs without headings. It would be better to mention themes as headings and discuss under them.

Response: Thank you for your feedback. We have added subheadings to the discussion section in order for it to read better. 

Comment 11: 

The authors have discussed well on the marriage and impact on education in a patriarchal society. However, this is missing information on how child marriage and education has influenced migrant women to self-management of diabetes. Please include this information to add richness to the finding of this study.

Response: Thank you for your comment. There is not enough published literature that make the connection between child marriage and education on migrant women’s diabetes self-management. We hope that our study adds to this topic. 

Comment 12: 

Please link how economic, systemic, informational, cultural, and linguistic barriers impact migrant women to manage diabetes.

Response: Thank you for your comments. We have added a sentence to link these barriers to the participants in our study. 

Comment 13: 

Please mention scientific evidence to support ‘As part of Nepali culture, people do not purposely engage in exercise activities to remain physically active; it is simply achieved through routine activities during daily life’.

Response: Thank you for point this out. We have added reference to support this sentence. 

Comment 14: 

Please mention what are these several studies, “Several studies conducted in Nepal associating depression with diabetes reported that low educational status and poor understanding of the disease was associated with poor mental health outcomes among diabetics”.

Response: Thank you for pointing this out. We have added reference to support this sentence.

Comments from Reviewer 2

Comment 1: 

First, refers the link with studies on patriarchy and overweight, as this is the main connection of the story. 

Response: Thank you for your comment. This study was based on a pilot project we conducted on diabetes management among the NSB community. We included the paper on excessive weight and poor dietary practices as one of the few papers related to the topic. However, we were unable to explore the association between weight and diabetes as it was beyond the scope of the study. 

Comment 2:

Second, provide slightly more practical implication in the discussion part, what are the policy implications of the results?

Response: Thank you for your feedback. We have added a few sentences in the conclusion section to add some policy implications from the findings of our study. 

We look forward to hearing from you in due time regarding our submission and to respond to any further questions and comments you may have. 

Sincerely,

Aditi Sharma

11/1/2021

---

## [Decision Letter · Decision Letter 1]

3 May 2022

The influence of patriarchy on Nepali-speaking Bhutanese women’s diabetes self-management

PONE-D-21-19730R1

Dear Dr. Aditi Sharma,

We’re pleased to inform you that your manuscript has been judged scientifically suitable for publication and will be formally accepted for publication once it meets all outstanding technical requirements.

Kind regards,

Sharon Mary Brownie

Academic Editor

PLOS ONE

Reviewers' comments:

Reviewer's Responses to Questions

**Comments to the Author**

1. If the authors have adequately addressed your comments raised in a previous round of review and you feel that this manuscript is now acceptable for publication, you may indicate that here to bypass the “Comments to the Author” section, enter your conflict of interest statement in the “Confidential to Editor” section, and submit your "Accept" recommendation.

Reviewer #1: All comments have been addressed

Reviewer #3: All comments have been addressed

2. Is the manuscript technically sound, and do the data support the conclusions?

Reviewer #1: Yes

Reviewer #3: Yes

3. Has the statistical analysis been performed appropriately and rigorously? 

Reviewer #1: Yes

Reviewer #3: Yes

4. Have the authors made all data underlying the findings in their manuscript fully available?

Reviewer #1: Yes

Reviewer #3: (No Response)

5. Is the manuscript presented in an intelligible fashion and written in standard English?

Reviewer #1: Yes

Reviewer #3: Yes

6. Review Comments to the Author

Reviewer #1: Dear Authors,

Thank you for revising manuscript based on input provided in earlier review. You may need to consider the journal requirements related to formatting. Your study will be valuable for future studies in migrant population.

Reviewer #3: Dear Authors,

I am glad you addressed all the reviewers' comments, adding details to make the study clearer. This study shows the influence of the patriarchy on the inadequacy of diabetes management in a rapidly growing Nepali-speaking Bhutanese community, which is subject very relevant. Understood the age was not the focus of this study, but agree with the comment 9 of the first reviewer, that it would be interesting to observe the differences among ages, as in some data you transcribed the patients referred that they rely on members of the family to help them with the translation in the consultations, that sometimes were females. Also because 5 of them had some access to education and I wondered whether they were among the 6 patients under 49 years old. At younger ages, the influence of the occidental culture and education could have some influence on the development of different policies among different groups. Maybe you could evaluate it better in a future study with a bigger sample.

Yours Sincerely

7. PLOS authors have the option to publish the peer review history of their article (what does this mean?). If published, this will include your full peer review and any attached files.

Reviewer #1: **Yes: **Mandira Adhikari

Reviewer #3: No

---

## [Editor Report · Acceptance letter]

16 Aug 2022

PONE-D-21-19730R1 

The influence of patriarchy on Nepali-speaking Bhutanese women’s diabetes self-management 

Dear Dr. Sharma:

I'm pleased to inform you that your manuscript has been deemed suitable for publication in PLOS ONE. Congratulations! Your manuscript is now with our production department. 

Kind regards, 

on behalf of

Professor Sharon Mary Brownie 

Academic Editor

PLOS ONE